# Nanofiber Scaffold Based on Polylactic Acid-Polycaprolactone for Anterior Cruciate Ligament Injury

**DOI:** 10.3390/polym14152983

**Published:** 2022-07-23

**Authors:** Rifqha Huriah, Dyah Hikmawati, Sofijan Hadi, Tahta Amrillah, Che Azurahanim Che Abdullah

**Affiliations:** 1Study Program of Physics, Department of Physics, Faculty of Science and Technology, Universitas Airlangga, Surabaya 60115, Indonesia; dyah-hikmawati@fst.unair.ac.id; 2Study Program of Biomedical Engineering, Department of Physics, Faculty of Science and Technology, Universitas Airlangga, Surabaya 60115, Indonesia; rifqha.huriah-2017@fst.unair.ac.id; 3Study Program of Chemistry, Department of Chemistry, Faculty of Science and Technology, Universitas Airlangga, Surabaya 60115, Indonesia; sofijan-h@fst.unair.ac.id; 4Study Program of Nanotechnology, Faculty of Advance Technology and Multidiscipline-Universitas Airlangga, Surabaya 60115, Indonesia; tahta.amrillah@ftmm.unair.ac.id; 5Nanomaterial Synthesis and Characterization Lab, Institute of Nanoscience and Nanotechnology, Universiti Putra Malaysia, Serdang 43400, Selangor, Malaysia; azurahanim@upm.edu.my; 6Department of Physics, Faculty of Science, Universiti Putra Malaysia, Serdang 43400, Selangor, Malaysia

**Keywords:** Anterior Cruciate Ligament (ACL) injury, scaffold nanofiber, Polylactic acid (PCL), Polycaprolactone (PCL)

## Abstract

Anterior Cruciate Ligament (ACL) injuries are becoming more prevalent in athletes. Anterior Cruciatum Ligament Reconstruction (ACLR) surgery was used to treat ACL injuries and resulted in a recurrence rate of 94% due to the biomechanically repaired tissue being weaker than the original tissue. As a result, biodegradable artificial ligaments must be developed that can withstand mechanical stress during neoligament formation and stabilize the ACL. The purpose of this study is to determine the effect of composition variations in polylactic acid (PLA) and polycaprolactone (PCL) used as ACL nanofiber scaffolds on ultimate tensile strength (UTS) and modulus of elasticity, fiber diameter, cytotoxicity level, and degradation level, as well as the PLA-PCL concentration that provides the best value as an ACL scaffold. Electrospinning was used to fabricate the nanofiber scaffold with the following PLA-PCL compositions: A (100:0), B (85:15), C (80:20), D (70:30), and E (0:100) (wt%). The functional group test revealed no new peaks in any of the samples, and the ester group could be identified in the C-O bond at wave numbers 1300–1100 cm^−1^ and in the C=O bond at wave numbers 1750–1730 cm^−1^. The average fiber diameter, as determined by SEM morphology, is between 1000 and 2000 nm. The unbraided sample had a UTS range of 1.578–4.387 MPa and an elastic modulus range of 8.351–141.901 MPa, respectively, whereas the braided sample had a range of 0.879–1.863 MPa and 2.739–4.746 MPa. The higher the PCL composition, the lower the percentage of viable cells and the faster the sample degrades. All samples had a cell viability percentage greater than 60%, and samples C, D, and E had a complete degradation period greater than six months. The ideal scaffold, Sample C, was composed of PLA-PCL 80:20 (wt%), had an average fiber diameter of 827 ± 271 nm, a living cell percentage of 97.416 ± 5.079, and a degradation time of approximately 219 days.

## 1. Introduction

Knee injuries, particularly to the Anterior Cruciatum Ligament (ACL), are a common occurrence in the modern era. According to US health data, over 100,000 medical procedures for ACL cases are performed each year, with an incidence rate of 1/3000 [1]. The ACL is a ligament found in the knee that connects the tibia and femur’s anterior bones [2]. The ACL’s primary function in human movement is to stabilize the knee by limiting anterior tibial translation and rotation [3]. A torn ACL cannot heal on its own; therefore, surgery to replace the ACL is required. Anterior Cruciatum Ligament Reconstruction (ACLR) surgery is used to treat ACL injuries. However, because the biomechanically repaired tissue is inferior to the original tissue, this surgery has a 94% recurrence rate. This is because the substituted material lacks the biomechanical properties of the natural ligament [4].

ACLR materials include those derived from the patient’s own tissue (autograft) and those obtained from donors (allograft) [5]. Clinically used autografts and allografts come from the hamstring tendon, semitendinosus quadriceps, and bone-patellar tendon-bone grafts (BPTB), with the BPTB graft integrating better than the other two materials [6]. The BPTB graft is made from a third of the ligament that connects the patella to the tibia. This autograft has the disadvantage of causing knee pain when the patient kneels. Meanwhile, AIAT allografts have a limited donor pool and a higher risk of infection transmission [6].

Artificial ligament research is currently focused on developing biodegradable biomaterials that can withstand mechanical stress during neo-ligament formation and restore the primary function of stabilizing the knee [7]. Biomaterials used to reconstruct the ACL must be biocompatible with high initial strength, biodegradable, capable of supplying sufficient nutrients and modulating signals to cells, and capable of promoting cellular adhesion and proliferation [8,9]. Current research focuses on degradable synthetic polymers that do not cause permanent foreign body reactions. Biodegradable synthetic polymers such as PLA, PCL, PGA, and PLGA have been developed recently as fibers, membranes, or patches for a variety of biomedical applications [9].

The materials listed above can be used to fabricate nanofiber scaffolds for ACL. PLA has an elasticity modulus of 0.35–3.5 Gpa [10]. PLA degrades over a period of ten months to four years [11]. PCL has a lower modulus of elasticity, ranging between 0.21 and 0.44 GPa, and a total degradation time of two years [12]. Thus, the PLA-PCL mixture has the potential to generate ACL with appropriate mechanical properties and a long degradation time [9].

On the other hand, computer-Aided Tissue Engineering was used to conduct research on the PLA-PCL scaffold. According to the simulation results, the composition variation of PLA-PCL 85:15 (wt %) has a % higher modulus of elasticity than the compositions 70:30 (wt %) and 97:3. (wt%). Due to the high likelihood of plastic deformation in the PLA-PCL 70:30 (wt%) composition, it is not suitable for ACLR [13]. The highest UTS value of 2.1 MPa and the highest elongation of 94.4% were obtained from a composition of PLA-PCL 80:20 (wt%) [9]. According to these two studies, appropriate techniques for designing ACL reconstruction materials are required. Thus, the purpose of this study is to investigate the mechanical properties, biocompatibility, and degradation time of a PLA-PCL fiber scaffold with a variety of compositions obtained via electrospinning and then manually braiding. It is important to note that the electrospinning method is known as the best method to fabricate micro membranes as well as a nanometric fiber size [14]. Electrospinning mainly involves the spinning process of a polymeric or biopolymeric solution forming a conjunction of randomly oriented fibers [14]. Our study offers a new pathway to obtaining a new combination of polymers, and it is hoped that an innovative product can be developed as a candidate material for ACL reconstruction with favorable mechanical properties, biocompatibility, and degradation time.

## 2. Materials and Methods

### 2.1. Materials

The primary ingredients, Poly-Lactic Acid (PLA) (Mn 100.000), and Poly-Caprolactone (PCL) (Mn 80.000) were purchased from Goodfellow and Sigma Aldrich, respectively. Chloroform and dimethylformamide were supplied by Merck (DMF). Oxoid supplied distilled water, Phosphate Buffered Saline (PBS), and 3-(4,5-dimethylthiazol-2-yl) 2,5-diphenyl tetrazolium bromide) (MTT).

### 2.2. Methods

Fabrication of nanofiber scaffolds from PLA-PCL was achieved using a Genlab HK-7 electrospinning type. The procedure begins by preparing a solvent from a 4:1 mixture of chloroform and DMF. After that, prepare a solution of each PLA and PCL at a concentration of 10% *w/v* [9]. The PLA-PCL sample was then dissolved in 10 mL of chloroform-DMF solution using a 200 rpm magnetic stirrer with modifications in the composition of the varied PLA-PCL, namely A (100:0), B (85:15), C (80:20), D (70:30), and E (0:100) in wt%. The dissolving process took two hours. After homogenizing the PLA-PCL, the resulting PLA-PCL copolymer appeared clear. Once the solution is homogeneous, the electrospinning process proceeded in order to create fiber. A syringe was filled with the homogeneous PLA-PCL solution. The syringe is then connected to the electrospinning apparatus. The electrospinning process employs a flat-shaped collector covered in aluminum foil to collect the fiber. The electrospinning parameters consist of the spinneret distance to the collector set to be 15 cm, the needle tip diameter being 0.5 mm, the flow rate being 2 mL/h, and the voltage being 20 kV [9]. The electrospun fiber was termed electrospun, whereas the single nanofiber was called nanofiber. The electrospun fiber was then cut to a size of 2 × 6 cm before being twisted and braided.

### 2.3. Characterization

#### 2.3.1. Functional Group Test Using Fourier Transform Infrared (FTIR) Spectrophotometer

Functional group was analyzed using FTIR (Shimadzu IRTracer-100) at wave number 400–4000 cm^−1^. The FTIR assessment was performed using a cuvette filled with 1 × 1 cm sample pieces. The FTIR method was used to evaluate the spectrum relationships between the percentage of transmission and the wavenumber (cm^−1^).

#### 2.3.2. Diameter Measurement of the PLA-PCL Nanofiber Scaffold with Scanning Electron Microscope (SEM)

The nanofiber scaffold was morphologically characterized using a Hitachi TM3000 Tabletop Microscope at magnifications of 5000× and 10,000×. The SEM image was further analyzed using the ImageJ application to determine the fiber diameter value. To evaluate the particle size in the SEM image using ImageJ, the first step is to calibrate the image pixel size against the reference size. Typically, the reference size is displayed on the SEM image as a line with a scale to indicate the magnification level used. The diameter of the average fiber can be determined using ImageJ software, (Version 1.50b, National Institutes of Health, Bethesda, MD, USA) and the data is then processed using Originlab 2016 to generate a sample diameter distribution plot. Another parameter that can be determined microscopically from the surface structure is the light-dark area fraction; the percentage of dark and light areas, where dark areas depict empty space and light areas represent the fiber formed, using the ImageJ application. To begin, the SEM image is segmented using threshold to make the distinction between nanofiber and background more obvious. After adjusting the threshold, the area fraction analysis can be performed using Image J’s histogram feature. The histogram indicates that a value of 0 represents a dark area and 255 represents a light area. The dark area represents the fiber’s porosity value.

#### 2.3.3. Mechanical Properties Measurement of PLA-PCL Nanofiber Scaffolds

Mechanical testing was conducted on a machine equipped with an electronic universal material testing system (AGS-X, Shimadzu, Co., Kyoto, Japan). According to American Standard Testing and Material (ASTM) type V, the sample was formed into a dogbone shape. The load-deformation curve was recorded and used to determine the ultimate failure load (N). The UTS was determined by the slope of the linear region of the load-deformation curve at the point of maximum load-to-failure. The tensile strength of each sample was determined three times without braiding and three times with braiding. The UTS value (σ) was calculated using Equation (1), and Young’s modulus or modulus of elasticity (E) was calculated using Equation (2). Equation (2) is also used to determine the elongation (2).
(1)$$\sigma =\frac{F}{A}$$
(2)$$E\;=\frac{\sigma }{\varepsilon }\;\left(\mathrm{MPa}\right)$$
with σ = stress (N/m^2^), F = Force (N), A = sample area (m^2^) and ε = strain (Δl/l_o_)

#### 2.3.4. Degradation Test of PLA-PCL Nanofiber Scaffold

The degradation test is used to determine how long a sample can remain viable in the body after being applied. The study used Phosphate Buffer Saline (PBS) as a buffer solution. Prior to immersing the nanofiber samples in PBS (Wo) solution, the weight of the samples was determined. Following that, samples were taken on the seventh, fourteenth, twenty-first, and twenty-eighth days following immersion. Prior to weighing, the sample was dried to determine its weight in the absence of any residual PBS (wt) solution. The weight loss will reveal the mass lost (Equation (3)), the rate of degradation, and the estimated time until the sample runs out.
(3)$$\mathrm{Mass}\;\mathrm{Loss}\left(\%\right)=\frac{\mathrm{Wo}\;\left(g\right)-\mathrm{Wt}\;\left(g\right)}{\mathrm{Wo}\;\left(g\right)}\times 100\%$$

#### 2.3.5. Cytotoxicity Test (MTT Assay) PLA-PCL Nanofiber Scaffold

Cytotoxicity testing was performed to ascertain the sample’s viability and to determine the level of toxicity in the PLA-PCL Nanofiber Scaffold sample. MTT assay is one of the cytotoxicity test methods. This test employed a 96-well microplate as the container for the sample testing. Following that, the cut samples were placed in plates and incubated for 24 h at 37 °C and 5% CO_2_. The MTT (3-(4,5-dimethyl-2-thiazolyl)-2,5-diphenyl-2H-tetrazolium bromide) reagent was then added and the mixture was incubated at 37 °C for another 4 h. Purple formazan crystals developed on the bottom after incubation, which were then dissolved using 200 uL DMSO for 30 min. Optical density was used to determine cell viability using an ELISA Multiplate Reader. The MTT assay results are denoted as a percentage of the total number of living cells, as defined by Equation (4).
(4)$$\%\;\mathrm{Live}\;\mathrm{cells}=\frac{\mathrm{OD}\;\mathrm{samples}-\mathrm{OD}\;\mathrm{medium}}{\mathrm{OD}\;\mathrm{cells}-\mathrm{OD}\;\mathrm{medium}}\times 100\%$$

## 3. Results

We have successfully fabricated a PLA-PCL nanofiber scaffold with a range of compositions, namely A (100:0), B (85:15), C (80:20), D (70:30), and E (0:100) in wt%. Electrospun nanofibers are white in color, which we collected on aluminum foil as illustrated in Figure 1a. The electrospun fibers are then separated from the aluminum foil. Following that, the sample was cut to a size of 2 × 6 cm and manually twisted and braided. The resulting sample is divided into two sections for tensile testing purposes, namely the braided sample (denoted by A) and the unbraided sample (denoted by B), as illustrated in Figure 1b.

The FTIR functional group analysis revealed that the PLA-PCL Nanofiber Scaffold sample possesses identical functional groups deriving from the ester group (O=C-O). Table 1 and Figure 2 represent the FTIR results. The FTIR functional group assessment demonstrates that the PLA and PCL mixture was well-formed. The FTIR analysis of the samples in Table 1 shows the presence of C-H stretching, C=O stretching, C-H bend, C-O stretching, and C-O-C bend bonds. Between 1300 and 1100 cm^−1^, the visible absorption band corresponds to the C-O stretching functional group, while 1750 to 1730 cm^−1^ corresponds to the C=O stretching functional group, which is characteristic of the PLA and PCL ester groups [15]. The C-O groups in PCL and the O-H hydroxyl groups in PLA serve as a marker for the interaction between the PLA and PCL molecules [16].

The morphological technique employs Scanning Electron Microscopy to determine the structure of a sample. 10,000× magnification was used in this measurement to ensure that the sample was clearly visible, and fiber diameter measurements were analyzed using ImageJ software. The results of the fiber diameter examination are listed in Table 2, and the SEM observations are shown in Figure 3. Due to the absence of beads, all samples exhibited perfect fiber formation morphologically. In Table 2, the fiber diameter decreases as the composition of PLA decreases, except for sample C. PLA has a higher molecular weight than PCL, which results in a more viscous solution than PCL. Thus, as the composition of PLA increases, the solution becomes thicker, increasing the diameter of the resulting fiber. The fiber formed from all samples is quite good, with an average diameter of 50–1000 nm, which is consistent with the diameter of the fiber in the extracellular matrix [17]. The distribution of fiber diameter appears to be non-uniform. This is due to a number of factors, including the voltage selected and the constant distance between the syringe and the collector for all samples. While the composition of the sample varies, the solution’s viscosity varies accordingly. To achieve a consistent diameter, it is necessary to combine several electrospinning process parameters, including the solution’s viscosity, the electric voltage, and the distance between the syringe and the collector. As illustrated in Figure 3, some fibers are fused (fiber fuse) to the fiber beneath. This fused fiber will act as the focal point for the occurrence of stress in the fiber, resulting in an uneven distribution of force across the fiber when a force is applied. As a result, the mechanical properties will be weakened [9].

Another parameter that can be determined microscopically from the surface structure is the fraction area, as illustrated in Figure 4. The percentage of dark and light areas can be calculated using the ImageJ software, where dark areas represent empty space and light areas represent formed fiber. The histogram (Figure 4) indicated that 0 represented a dark area (fiber porosity) and 255 represented a light area (fiber density). The area fraction analysis performed using the Histogram ImageJ feature (Figure 4) revealed that the dark area represents the fiber porosity value and the bright area represents the fiber density value. The dark area represents the fiber porosity value and the bright area represents the fiber density value. The greater the density of the fiber, the more robust the mechanical properties produced. The lower the fraction of dark areas representing empty space, the greater the increase in fiber density. The high fiber density contributed significantly to the scaffold’s mechanical strength. The sample with the highest nanofiber density was B, with an 85:15 (wt%) composition of PLA and PCL.

Tensile tests were used to determine the sample’s tensile strength and modulus of elasticity. The samples that pass this tensile test should exhibit the same biomechanical behavior as ACL tissue in terms of UTS values and modulus of elasticity. The sample is divided into two groups in this test: unbraided and braided samples. The tensile strength test results are listed in Table 3 and graphed in Figure 5. According to Table 3 and Figure 5, the tensile strength test performed on both the unbraided and braided samples revealed that the braiding process did not significantly increase the UTS value in the sample. The unbraided PLA-PCL sample has a UTS value of approximately 0.7–2.1 MPa, which decreases as PCL composition is added [9]. This result is nearly identical to that of the braided sample in Table 3.

The degradation test is used to determine the maximum duration of the sample in the body following implantation. The liquid used is Phosphate Buffer Saline (PBS), which is a salt solution that is similar to human body fluids. The test results are presented in Figure 6 as a percentage of mass lost due to degradation after 7, 14, 21, and 28 days. On the basis of Figure 6, Table 4 presents the degradation rate and estimates of when the sample will be completely degraded. According to Table 4, the estimated time for complete degradation of sample A is approximately 104 days. Sample B will degrade completely in approximately 176 days, whereas sample C will degrade completely in approximately 219 days. Sample D will undergo 354 days of degradation, while sample E will undergo 534 days of degradation. Due to the higher caprolactone content in Sample E, it degraded more slowly than the other four samples. The estimated time required for the sample to degrade completely can be predicted using the regression equation from Figure 6. The material used in ACL reconstruction must not degrade faster than the postoperative recovery process. ACL reconstruction typically takes at least six months to restore a functional knee [18].

The MTT assay is used to determine cytotoxicity in Baby Hamster Kidney (BHK-21) cells. This test is performed by calculating the percentage of viable cells. Figure 7 illustrates the cytotoxicity results. According to Figure 7, the cell viability values decreased with increasing PCL composition in all samples. This is due to the decomposition of methylene, which results in a decrease in the acidic pH, which can be harmful to cells [19]. As illustrated in Figure 7, all samples of PLA-PCL nanofiber scaffold have a cell viability percentage greater than 60%, indicating that the sample is not toxic [20]. The ACL is a knee ligament. Along with the ACL, the knee is stabilized by three additional ligaments: the posterior cruciate ligament, the lateral (outer side), and the medial (inner side) collateral ligaments. The ACL is a complex structure that connects the medial wall of the lateral femoral condyle to the middle region of the tibial plateau. It is critical for knee stability [21]. Ligaments, in conjunction with the medial and lateral meniscus of the leg muscles, work to stabilize the joint and provide strength for knee activity [5].

## 4. Discussion

The ACL is made up of two ribbon-like fibers known as the anteromedial and posterolateral bands. The anteromedial band is particularly tight during flexion and extension, and it becomes even more so when the knee is bent. When the knee is bent, the posterolateral bands become moderately relaxed. The ACL is made up of several collagen fibers that are intertwined, giving it a high tensile strength. The ACL angle varies between 60–80° in the tibial and femoral basins [11].

To allow for degradation of properly bearing specimens, greater UTS and modulus of elasticity are required. Males had UTS ACL values ranging from 16.3–36.4 MPa, while females had values ranging from 13.7–31.5 MPa. Meanwhile, men’s modulus of elasticity ranges between 93 and 163 MPa, while women’s modulus of elasticity ranges between 49 and 149 MPa [22]. The ACL measures 32.5 ± 2.6 mm in length and approximately 7–11 mm in width [23].

Due to their superior mechanical properties and biodegradability, PLA and PCL are promising materials for use as a substitute for the ACL. PLA is biodegradable, stiff, and tough, but has a low toughness. PLA has a tensile yield strength of 48–110 MPa and a tensile elongation of 2.5–100%. PLA melts at 153 °C and has a glass transition temperature of 60 °C [24]. PCL is a partially crystalline synthetic polyester with a low melting point of 56–65 °C and a glass transition temperature of −60 °C. PCL is easily mixed with other materials to improve its mechanical properties due to its low melting point. PCL has a tensile strength of 4–785 MPa and an elongation range of 20–1000% [25].

As a result of this research, the PLA-PCL nanofiber scaffold has a range of fiber diameters without beads. Fibers are considered adequate when their average diameter is between 50 and 1000 nm, which corresponds to the diameter of the fiber in the extracellular matrix [17]. Combining PLA and PCL results in fibers with an average diameter of 480–660 nm [26]. Because nanometer-sized fibers have a high surface area ratio, they can accommodate more cells than other structures [27].

The braided sample’s mechanical properties indicated that the UTS value was not comparatively higher than the unbraided sample. The braiding process should improve the scaffold’s mechanical properties. The orientation of the fibers formed also has an effect on the mechanical properties of the scaffold. Random fiber orientation results in a lower UTS value than parallel fiber orientation of nanofiber [28]. In a study conducted by Sharma and team, non-braiding samples had UTS values ranging between 0.7 and 2.1 MPa, but the value decreased when PCL composition was added. This result is nearly identical to the unbraiding sample (Figure 5). According to Figure 5, the UTS value and modulus of elasticity remain outside the optimal range of ACL values. It is necessary to conduct research by varying the concentrations of PLA and PCL solutions, leveraging a drum-shaped electrospinning collector to generate nanofibers with parallel fiber orientation, and employing torsion and braiding tools to achieve the best results.

The PLA-PCL scaffold used in this study qualified as an ACL scaffold due to its biodegradability and cytotoxicity. The greater the proportion of PCL in a sample, the slower the degradation rate. PCL has hydrophobic properties, which contribute to its slow degradation rate [29]. ACL reconstruction typically takes at least six months to complete [18]. All samples except sample A, which contained only PLA, met the standard time in this study. Increased PCL composition decreases cell viability as a result of the methylene decomposition process, which results in an acidic pH that can damage cells [19]. However, all samples of PLA-PCL nanofiber scaffolds demonstrated a cell viability percentage greater than 60%, indicating that the samples were not toxic [20].

## 5. Conclusions

Variations in the composition of PLA:PCL resulted in the formation of distinct functional groups for each PLA and PCL in this study. According to the results of morphology data obtained from SEM, all samples have fibers with a diameter less than 1000 nm. A DSC and XRD measurements will be necessary for the future to understand whether the samples are crystalline or amorphous which is important to understanding their mechanical properties. The Tensile Test, UTS, and Modulus of Elasticity values for both unbraided and braided samples remained within the original ACL range, and the higher the PCL composition, the lower the UTS and Modulus of Elasticity values. According to the cytotoxicity test, all samples had a percentage of living cells greater than 60%, indicating that the sample was non-toxic. Additionally, the higher the PCL composition, the lower the percentage of viable cells and slower the sample’s degradation. The optimal composition of PLA-PCL for ACL scaffold tissue engineering is 80:20. This compositional variation resulted in an ideal scaffold sample with an average fiber diameter of 827 ± 271 nm, a percentage of living cells of 97.416 ± 5.079, and a degradation time of approximately 219 days.

## Figures and Tables

**Figure 1 polymers-14-02983-f001:**
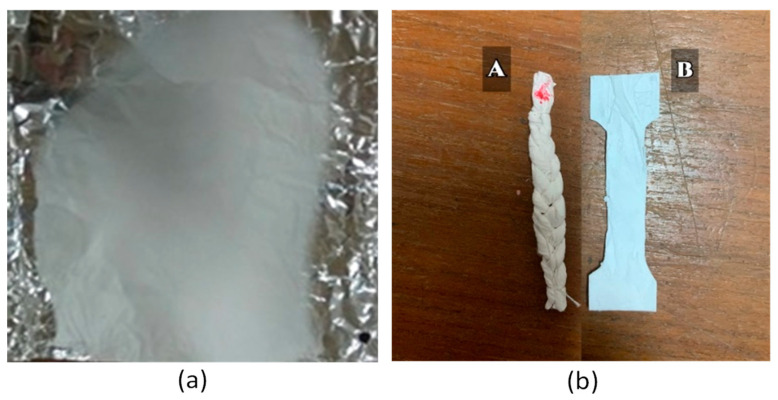
(**a**) Nanofiber sample from electrospinning. (**b**) Braided (A) and unbraided (B) samples.

**Figure 2 polymers-14-02983-f002:**
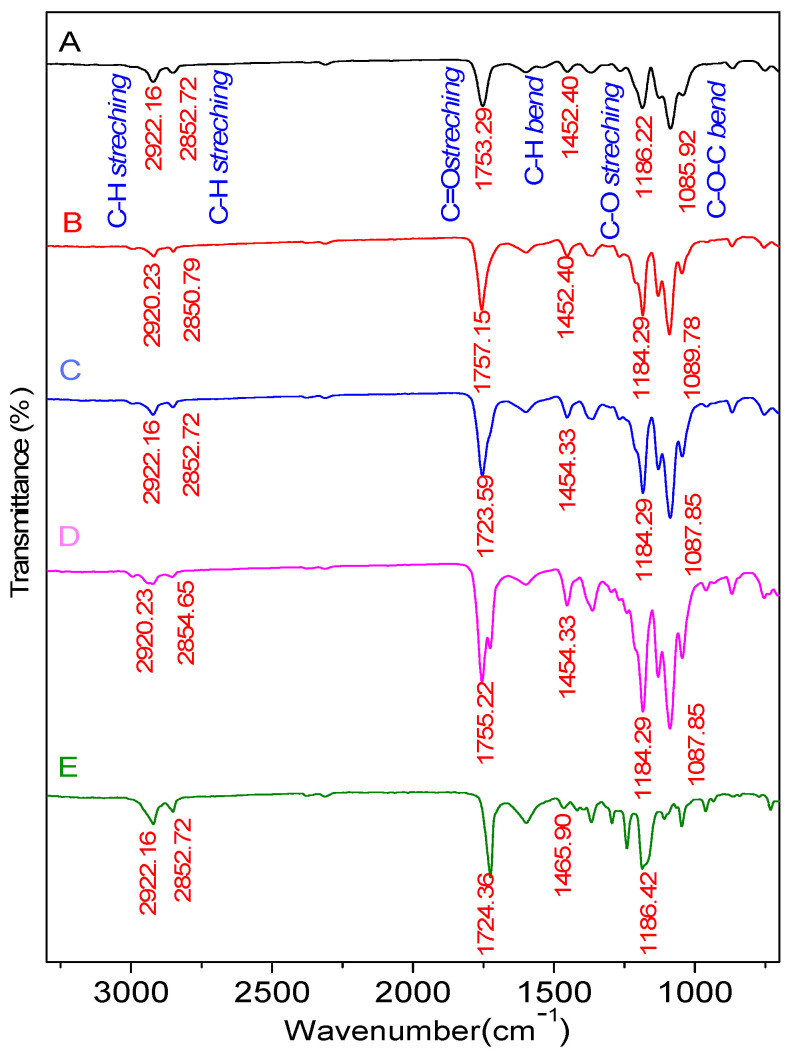
FTIR Spectrum of samples with composition of PLA:PCL; A (100:0), B (85:15), C (80:20), D (70:30), and E (0:100) in wt%.

**Figure 3 polymers-14-02983-f003:**
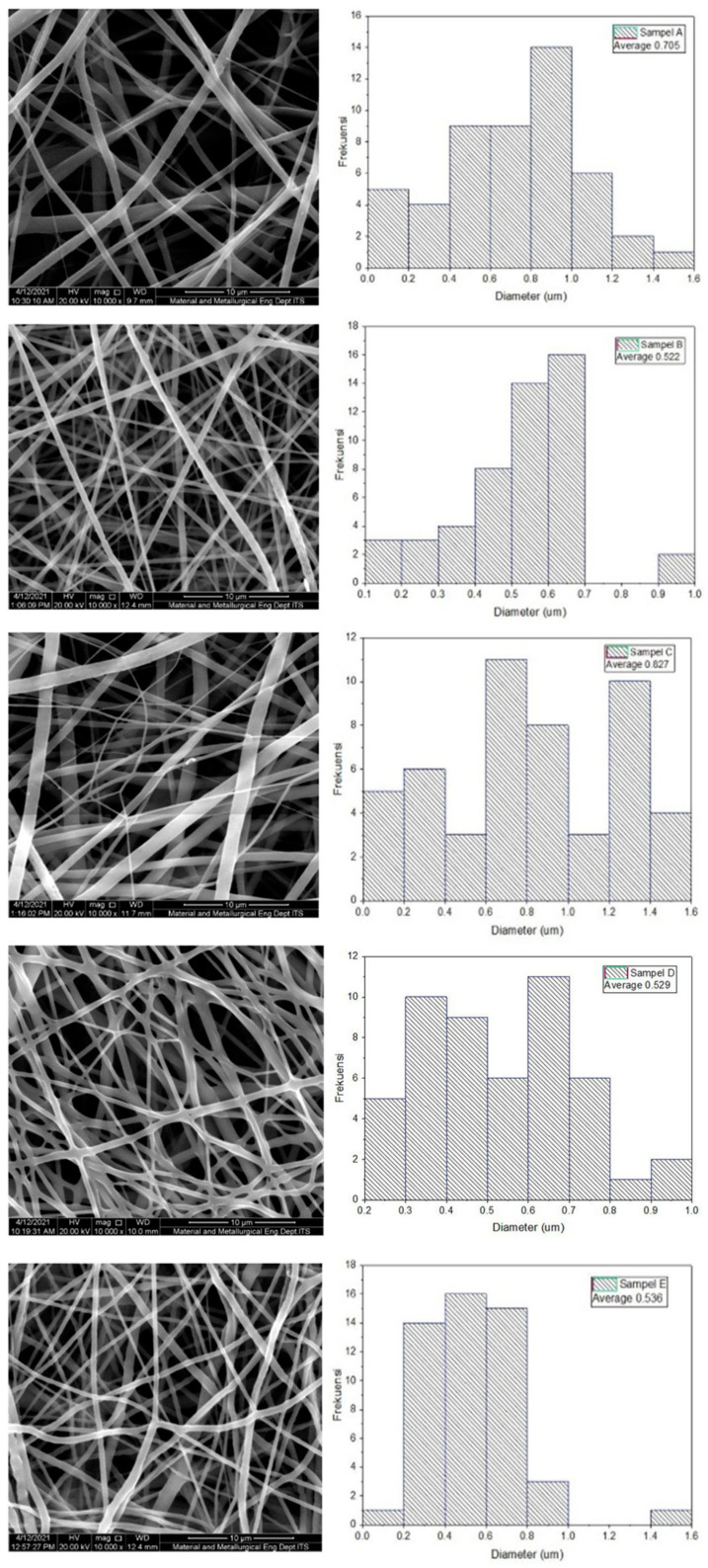
SEM observation results and fiber diameter distribution of the samples with a composition of PLA:PCL; A (100:0), B (85:15), C (80:20), D (70:30), and E (0:100) in wt%.

**Figure 4 polymers-14-02983-f004:**
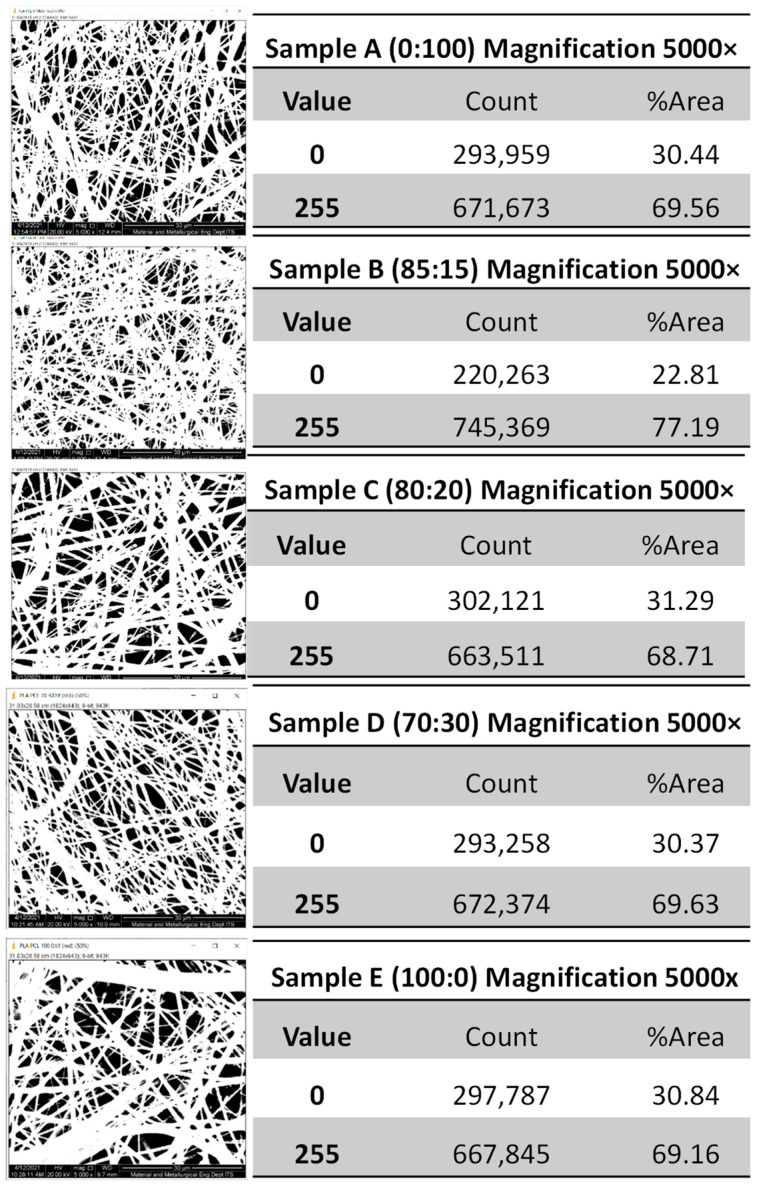
Image J histogram, light-dark areas of PLA-PCL nanofiber scaffolds of the samples with a composition of PLA:PCL; A (100:0), B (85:15), C (80:20), D (70:30), and E (0:100) in wt%.

**Figure 5 polymers-14-02983-f005:**
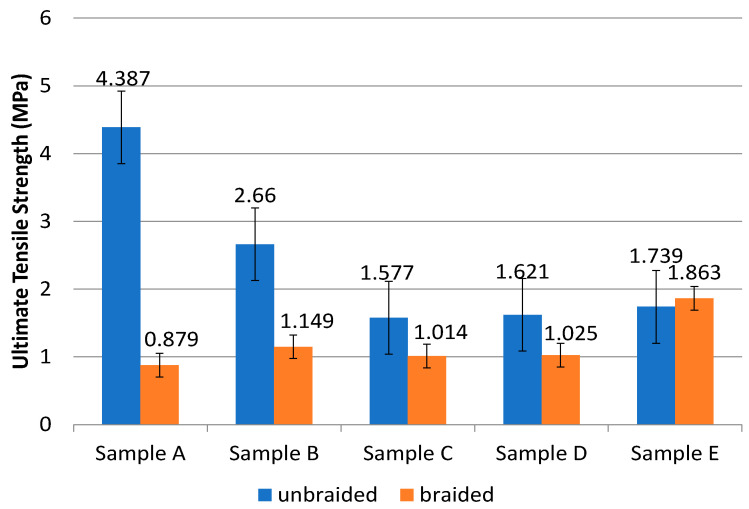
Graph of ultimate tensile strength (UTS) value of the samples with a composition of PLA:PCL; A (100:0), B (85:15), C (80:20), D (70:30), and E (0:100) in wt%.

**Figure 6 polymers-14-02983-f006:**
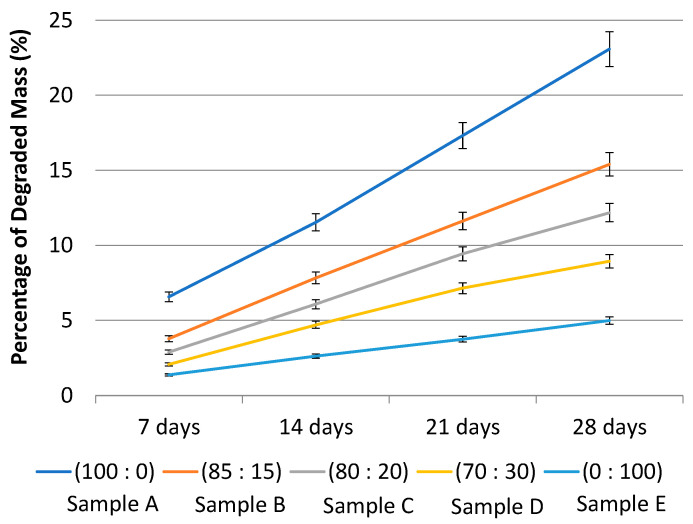
Graph of the average percentage of degraded mass of the samples with a composition of PLA:PCL; A (100:0), B (85:15), C (80:20), D (70:30), and E (0:100) in wt%.

**Figure 7 polymers-14-02983-f007:**
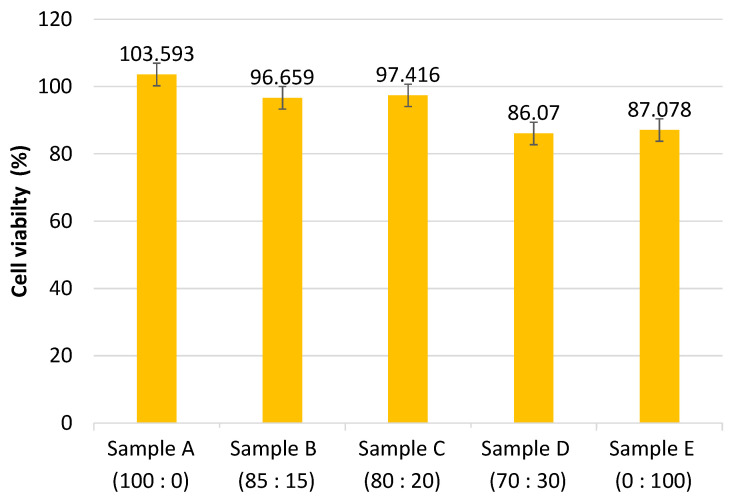
Graph of cell viability percentage of the samples with a composition of PLA:PCL; A (100:0), B (85:15), C (80:20), D (70:30), and E (0:100) in wt%.

**Table 1 polymers-14-02983-t001:** FTIR data showing the functional groups of nanofiber scaffolds.

Bond	Standard Value	$\mathrm{Wavenumber}\;({\mathrm{cm}}^{-1})$				
		A PLA-PCL (100:0)	B PLA-PCL (85:15)	C PLA-PCL (80:20)	D PLA-PCL (70:30)	E PLA-PCL (0:100)
**C-O stretch**	1300–1100	1186.22	1184.29	1184.29	1184.29	1186.22
**C-O-C bend**	1150–1050	1085.92	1089.78	1087.85	1087.85	-
**C-H bend**	1485–1445	1452.40	1452.40	1454.33	1454.33	1465.90
**C=O stretch**	1750–1730	1753.29	1757.15	1753.29	1755.22	1724.36
**C-H_2_** **stretch**	2935–2915	2922.16	2920.23	2922.16	2920.23	2922.16
**O-H stretch**	3570–3200	-	-	-	-	3525.88

**Table 2 polymers-14-02983-t002:** Compilation of fiber diameter of the samples.

Sample	PLA-PCL Composition (wt%)	Minimum Diameter (nm)	Maximum Diameter (nm)	Average Diameter (nm)
**A**	100:0	108	1.528	705 ± 328
**B**	85:15	149	977	522 ± 192
**C**	80:20	155	1.510	827 ± 271
**D**	70:30	248	957	529 ± 104
**E**	0:100	195	1.412	536 ± 154

**Table 3 polymers-14-02983-t003:** The results of tensile strength of the samples.

Sample	Composition of PLA:PCL (wt%)	Unbraided		Braided	
		UTS (MPa)	Modulus of Elasticity (MPa)	UTS (MPa)	Modulus of Elasticity (MPa)
**A**	100:0	4.387 ± 1.90	141.901 ± 36.96	0.879 ± 0.15	4.523 ± 1.32
**B**	85:15	2.660 ± 0.35	121.373 ± 25.39	1.149 ± 0.28	2.739 ± 1.24
**C**	80:20	1.578 ± 0.37	93.698 ± 31.53	1.014 ± 0.48	4.746 ± 2.51
**D**	70:30	1.621 ± 0.05	76.908 ± 5.83	1.025 ± 0.31	3.095 ± 1.29
**E**	0:100	1.739 ± 1.30	8.351 ± 2.10	1.863 ± 0.45	3.042 ± 1.35

**Table 4 polymers-14-02983-t004:** Data of estimated time for sample to totally degrade.

Sample	Composition of PLA-PCL (wt%)	Degradation Rate (g/day)	Estimated Sample Time Out (days)
**A**	100:0	$2.0\;\times \;10^{-4}$	104
**B**	85:15	$1.4\;\times \;10^{-4}$	176
**C**	80:20	$1.0\;\times \;10^{-4}$	219
**D**	70:30	$1.0\;\times \;10^{-4}$	354
**E**	0:100	$5.0\;\times \;10^{-5}$	534

## Data Availability

Not applicable.

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
