# Peer review of "Nanofiber Scaffold Based on Polylactic Acid-Polycaprolactone for Anterior Cruciate Ligament Injury"

_polymers, 2022, doi:10.3390/polym14152983_

Round 1
Reviewer 1 Report
This paper reports an investigation about the fabrication of nanofibrous scaffolds from PLA and PCL combinations. The paper includes interesting results with suitable data analysis and discussion. This manuscript may be recommended for publication in Polymers after major revision indicated below.
INTRODUCTION
- Include the main novelty of the work.
- Improve the introduction section including a paragraph about electrospinning. Here there are some references authors could use:
https://doi.org/10.1007/s10704-020-00460-4
https://doi.org/10.1002/app.50893
https://doi.org/10.3390/polym13091501
https://doi.org/10.3390/polym14040665
https://doi.org/10.1002/jbm.a.37156
RESULTS
- How could the authors separate the electrospun membranes from the aluminum foil after the electrospinning process?
- In lines 308-309 “while women's modulus of elasticity ranges between 49 308 and 149 MPa [21]. The ACL measures 32.5 ± 2.6 mm in length and approximately 7-11 mm 309 in width.22” there are two different ways to insert a reference citation. Please unify the format of the reference citation following the Guide for Authors.
CONCLUSIONS
- In the Conclusions section, mention possible further studies in which these analyses may be useful.
REFERENCES
- Improve the format of the references section using the Guide for Authors
- Include more references of the journal.
Author Response
Review 1
This paper reports an investigation about the fabrication of nanofibrous scaffolds from PLA and PCL combinations. The paper includes interesting results with suitable data analysis and discussion. This manuscript may be recommended for publication in Polymers after major revision indicated below.
- INTRODUCTION
- Include the main novelty of the work.
- Improve the introduction section including a paragraph about electrospinning. Here there are some references authors could use: https://doi.org/10.1007/s10704-020-00460-4, https://doi.org/10.1002/app.50893, https://doi.org/10.3390/polym13091501, https://doi.org/10.3390/polym14040665, https://doi.org/10.1002/jbm.a.37156
Response: Thank you for the suggestion. We agree to improve the introduction part by adding the explanation of the novelty of the work and the suggested references. Please see the revised manuscript.
- RESULTS
- How could the authors separate the electrospun membranes from the aluminum foil after the electrospinning process?
- In lines 308-309 “while women's modulus of elasticity ranges between 49 308 and 149 MPa [21]. The ACL measures 32.5 ± 2.6 mm in length and approximately 7-11 mm 309 in width.22” there are two different ways to insert a reference citation. Please unify the format of the reference citation following the Guide for Authors.
Response: The electrospun membranes was separated from aluminum foil by mechanical exfoliation.
In the results part, we carefully check it again, especially for the typos and we have fixed it. Please see the revised manuscript.
- CONCLUSIONS
- In the Conclusions section, mention possible further studies in which these analyses may be useful.
Response: Thank you for the suggestions. We have added the explanation about the further study of our work.
- REFERENCES
- Improve the format of the references section using the Guide for Authors
- Include more references of the journal.
Response: We thanks for this suggestion. We apologize for the mistakes with the references. Therefore, we fixed and improved the references as shown in revised manuscript.

Reviewer 2 Report
The article is relevant to the polymers magazine. However, it does require major changes. The characteristics of the PLA and PCL used are incomplete. There is a lack of polydispersity of the polymers used. What kind of polylactide was used? Was it a homopolymer? Are you a heteropolymer? The article lacks research on the crystallinity of the system. There are no DSC measurements, it is not known whether the authors obtained crystalline or amorphous fibres. This can have a significant effect on mechanical properties. The authors write about the use of nonwovens for ACL. And there is no other data on the mechanical properties, such as the relative elongation at break. No error bars when measuring fibre diameters. No measurements of the contact angle of the obtained nonwoven fabrics. The result for the control is missing in the cell studies. The literature is not properly formatted.
Author Response
We thanks for the insight. We understand that we should have several data you required. However, due to limited time to complete the requested additional data, we cannot added it in our manuscript. Nevertheless, we have made some modification and improve the manuscript, especially the literature format. Please see the revised manuscript.

Round 2
Reviewer 1 Report
Authors carried out all the changes suggested during the previous review.
Reviewer 2 Report
The article is ready for publication. Congratulations